# Study of Comminution Kinetics in an Electrofragmentation Lab-Scale Device

Angel R. Llera [1], Ana Díaz [1], Francisco J. Pedrayes [2], Juan M. Menéndez-Aguado [1,*] and Manuel G. Melero [2]

[1] Escuela Politécnica de Mieres, University of Oviedo, c/Gonzalo Gutiérrez Quirós, 33600 Mieres, Spain; uo46888@uniovi.es (A.R.L.); diazdana@uniovi.es (A.D.)

[2] Departamento de Ingeniería Eléctrica, Electrónica, de Computadores y Sistemas, University of Oviedo, 33204 Gijón, Spain; pedrayesjoaquin@uniovi.es (F.J.P.); melero@uniovi.es (M.G.M.)

* Correspondence: maguado@uniovi.es; Tel.: +34-985-458-033

**Abstract:** A significant challenge in mineral raw materials comminution is the improvement of process energy efficiency. Conventional comminution techniques, although globally used, are far from being considered power-efficient. The use of high-voltage electric pulses in comminution is a concept that is worthy of study; despite its lack of industrial-scale validation after several decades of lab-scale research, it seems promising as a pretreatment leading to energy savings. In this article, the Cumulative Kinetic Model methodology is adapted to model the comminution effect in an electrofragmentation device, and study a dunite rock ore. The results show that product particle size distribution (PSD) can be predicted with reasonable accuracy using the proposed model.

**Keywords:** electrofragmentation; comminution; Marx generator; modeling

## 1. Introduction

Comminution operations are essential in mineral raw materials industries, and estimations of their share in global energy consumption range from 3 to 5% [1–4], so the improvement of process energy efficiency poses a significant challenge in mineral processing technology. Conventional comminution techniques, although globally used, are far from being considered power-efficient. The use of high-voltage electric pulses (HVEP) in comminution is a concept worth studying; despite its lack of industrial-scale validation after several decades of lab-scale research, it seems promising as a pretreatment leading to energy savings. Moreover, it is probably the only known comminution technology capable of maintaining its efficiency in a zero-gravity environment.

Initial research into HVEP use in comminution started in the mid-20th century to produce rock weakening and selective mineral fragmentation [5,6]. Some studies performed comparisons with conventional technologies on such issues as size reduction capability and energy consumption [7–10], while other studies focused on improving mineral liberation [11–15].

This study proposes a mathematical model to predict product PSD in an HVEP device after one or more electric pulses under specific working conditions. Preliminary tests showed the particular influence of pulse polarity on breakage results, so this effect will also be analyzed.

## 2. Experimental

### 2.1. Materials

Samples were supplied by the mineral processing plant at Mina David (Pasek Minerales), located in Landoi (Spain). This is the only dunite producer in Spain; despite the olivine content being too low (20–30%) to classify it as a dunite rock, it keeps this commercial denomination. Along with olivine, it is usually accompanied by orthopyroxene (8–16%), amphibole (14–20%) and chrysotile (0–33%). Moreover, other minerals can appear

in the open pit due to hydrothermal alterations, such as chlorite, serpentinite and clay group minerals. Table 1 shows the X-ray fluorescence (XRF) results. Further characterizations of this ore can be found in [16].

**Table 1.** XRF ore results (%) (L.O.I. = lost on ignition).

| SiO$_2$ | Al$_2$O$_3$ | Fe$_2$O$_3$ | MgO | CaO | K$_2$O | Others | L.O.I. |
|---|---|---|---|---|---|---|---|
| 39.86 | 3.00 | 7.62 | 35.34 | 1.73 | 0.07 | 0.35 | 11.91 |

Due to the high Mg content shown above, Pasek Minerales is currently developing an extraction process, aimed at producing high-quality magnesium oxide from dunite fines; any step towards a reduction in the specific energy consumption in the fines production process would be desirable.

To provide comminution characterization, a Bond ball mill standard test was performed on a representative sample, with a result of 11.6 kWh/t at 100 microns.

A sufficient amount of sample was prepared within narrow size intervals via sieving. These fractions can be considered monosizes, and they were tested separately to determine the influence of particle size. The selected intervals were (in microns): 5000/3350; 3350/2000; 2000/1000; 1000/500; 500/125 and 125/0. Table 2 shows the total weights of each monosize after sieving. Aliquots of 500 g were prepared for each monosize using a Jones sample divider (RETSCH, Haan, Germany).

**Table 2.** Sample weight after preparation.

| Monosize (μm) | Weight (kg) |
|---|---|
| 125/0 | 15.2 |
| 500/125 | 23.12 |
| 1000/500 | 18.34 |
| 2000/1000 | 24.46 |
| 3350/2000 | 12.34 |
| 5000/3350 | 18.30 |

*2.2. Methods*

2.2.1. HVEP Test Rig

The test rig (see Figure 1)is based on a Marx pulse generator SGSA 400-20 (HAEFELY, Basel, Switzerland), located at the Electrical Engineering Department facilities in Gijon (University of Oviedo, Spain). The main characteristics of this HVEP test rig are depicted in Table 3.

**Table 3.** General specifications of the HVEP generator.

| Parameter (unit) | Value |
|---|---|
| Maximum voltage (kV) | 400 |
| Maximum energy discharge (kJ) | 20 |
| Number of stages | 4 |
| Capacity/stage (μF) | 1 |

Figure 2 shows the diagram of a Marx impulse generator. The depicted C and Cs correspond to the test cell and the impulse capacitance, respectively. Rs and Rp are the resistances that define the pulse leading edge time and trailing edge time, respectively. The element SF represents the spark gap that starts the discharge of the impulse capacitance into the test cell, thus generating the requested pulse.

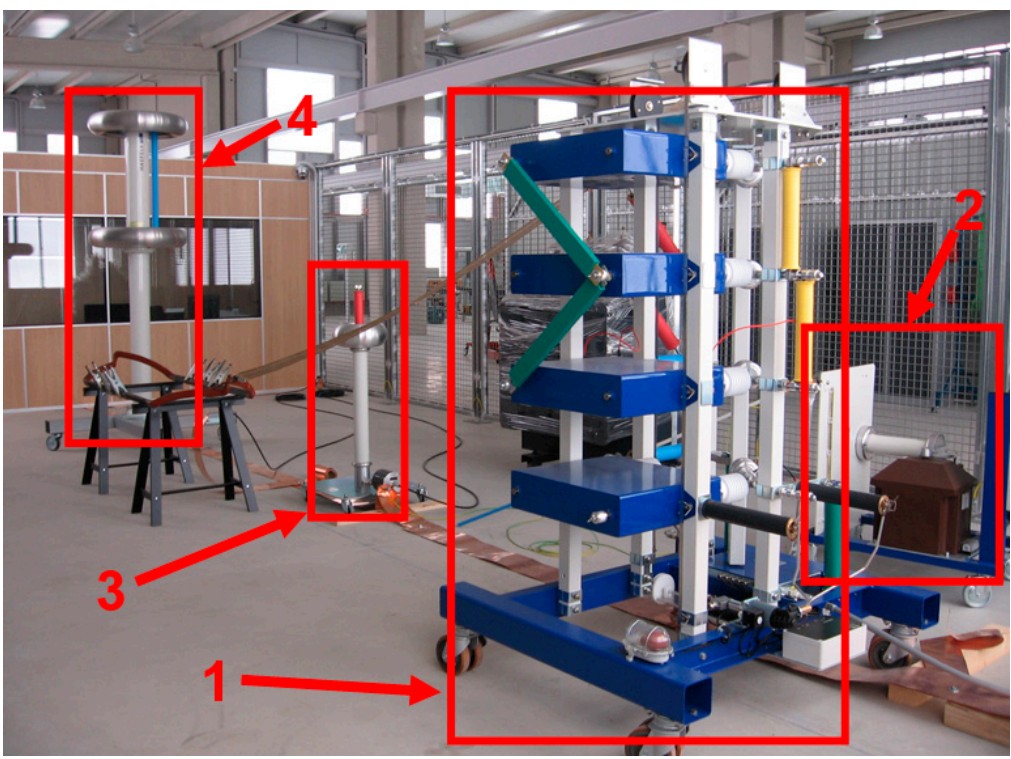

**Figure 1.** Test rig (1) pulse generator, (2) charge unit, (3) capacitive divider, (4) compensation circuit.

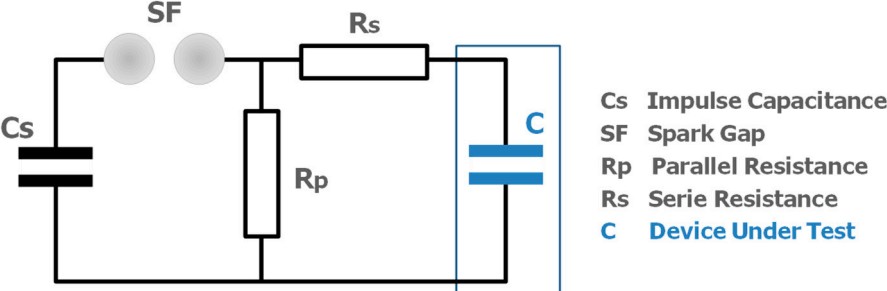

| | |
|---|---|
| Cs | **Impulse Capacitance** |
| SF | **Spark Gap** |
| Rp | **Parallel Resistance** |
| Rs | **Serie Resistance** |
| C | **Device Under Test** |

**Figure 2.** Marx generator diagram.

The pulse generation and measurement process are represented in Figure 3. Firstly, the desired impulse specifications are set in the generator control unit, including the number of work stages, peak impulse value, capacitance charging time and impulse polarity. Secondly, the charging unit raises the voltage to the specified peak value and the charging rectifier converts this to direct current, which is used to charge the generator capacitors. Afterwards, when the capacitors reach the pre-set voltage, the control unit orders the impulse to discharge on the sample within the test cell. Finally, the impulse is registered using a voltage divider in parallel, which permits the signal's digitalization and treatment.

In contrast to the devices used in previous studies [17–20], this test rig has the option of changing impulse polarity. This feature can be achieved by changing the positions of the charge unit diodes (Figure 4), inverting the voltage discharge polarity and thus getting positive or negative discharge impulses on the test sample. Figure 5 shows two examples of no-load impulse curves of different polarities (X axis time in microseconds; Y axis voltage in kV).

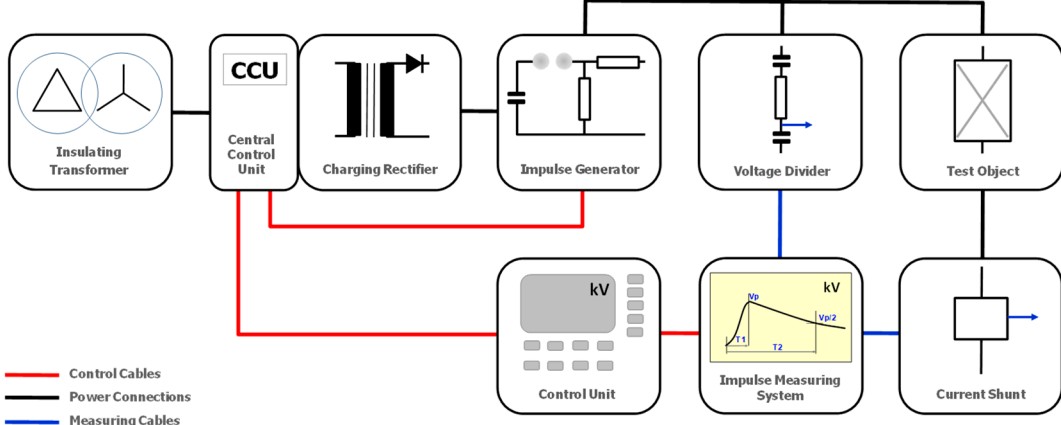

**Figure 3.** HVEP test rig block diagram.

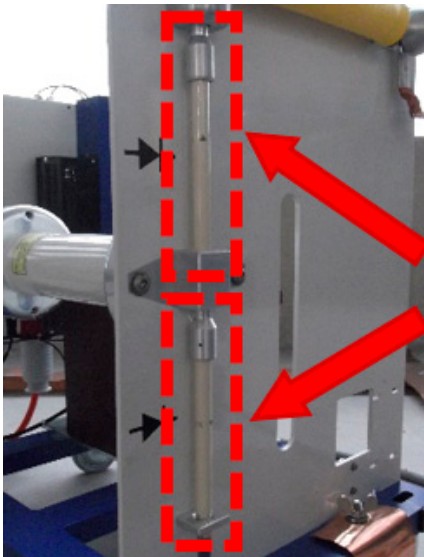

**Figure 4.** Charge unit diodes.

A relevant parameter in the electrofragmentation tests is the pulse rise time, for this must be short enough to produce a successful fragmentation [21]. Impulse discharge through a mineral sample requires enough voltage to overcome the sample dielectric strength, but the voltage achieved should not surpass the surrounding material's dielectric strength, because, in that case, the discharge would concentrate in the surrounding medium. Additionally, if a medium with higher electric permittivity surrounds the mineral sample, a very uneven distribution of the applied electric field occurs, with a high concentration in the mineral and a much lower concentration in the surrounding medium.

Both effects can be achieved by soaking the mineral sample in distilled water; at a very short pulse rise time, water's dielectric strength and permittivity are higher than rock's [21,22], as shown in Figure 6, which shows that the pulse rise time should be less than 500 ns.

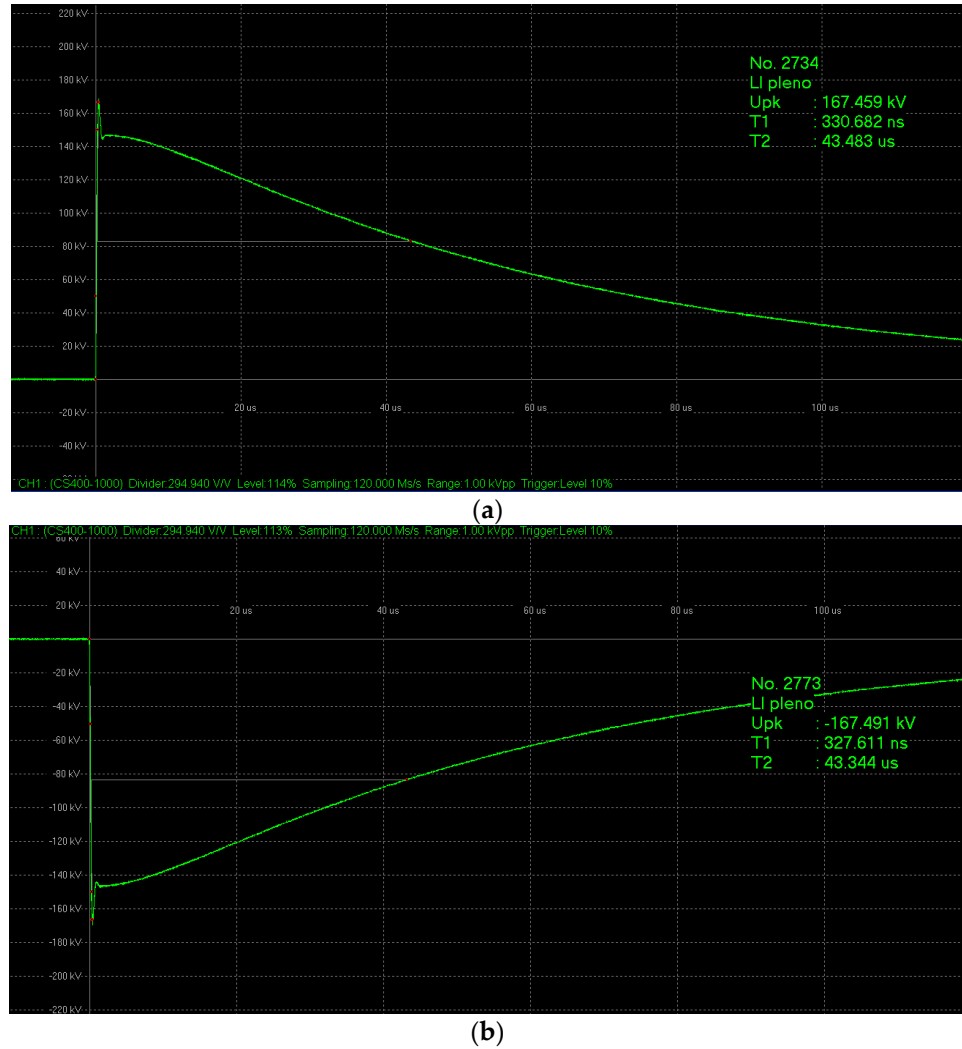

**Figure 5.** No-load impulses: (**a**) positive polarity; (**b**) negative polarity.

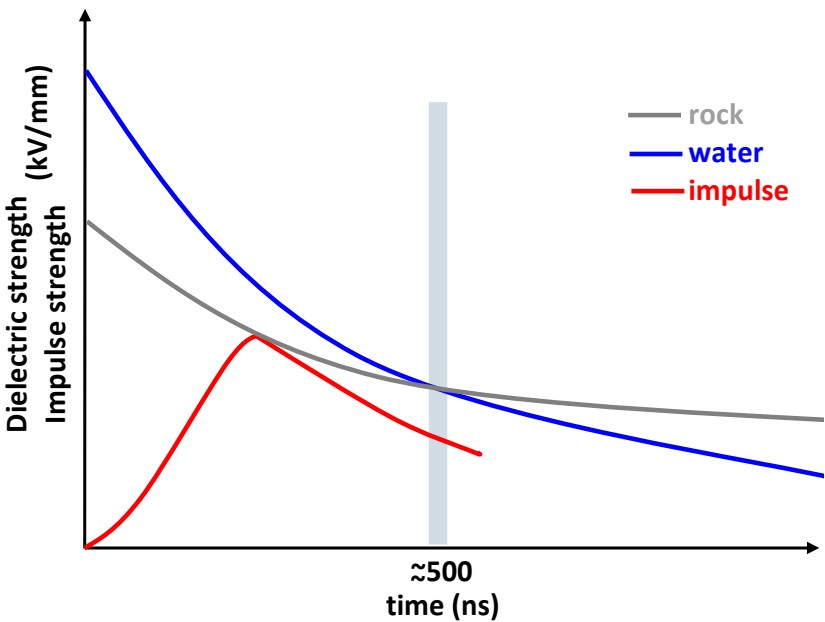

**Figure 6.** Variation of dielectric strength with the pulse rise time.

With the aim of a more significant reduction in the pulse rise time, we substituted the resistance Rs (Figure 2) for a short-circuit; thus, a pulse rise time around 300 ns can be achieved, with a peak voltage of 150 kV (this value was set in all tests performed), plus an additional value due to overshooting. Under these conditions, the discharge effect will concentrate in the mineral sample; the wave shapes obtained when applying these pulses (both with positive and negative polarity) are shown in Figure 7.

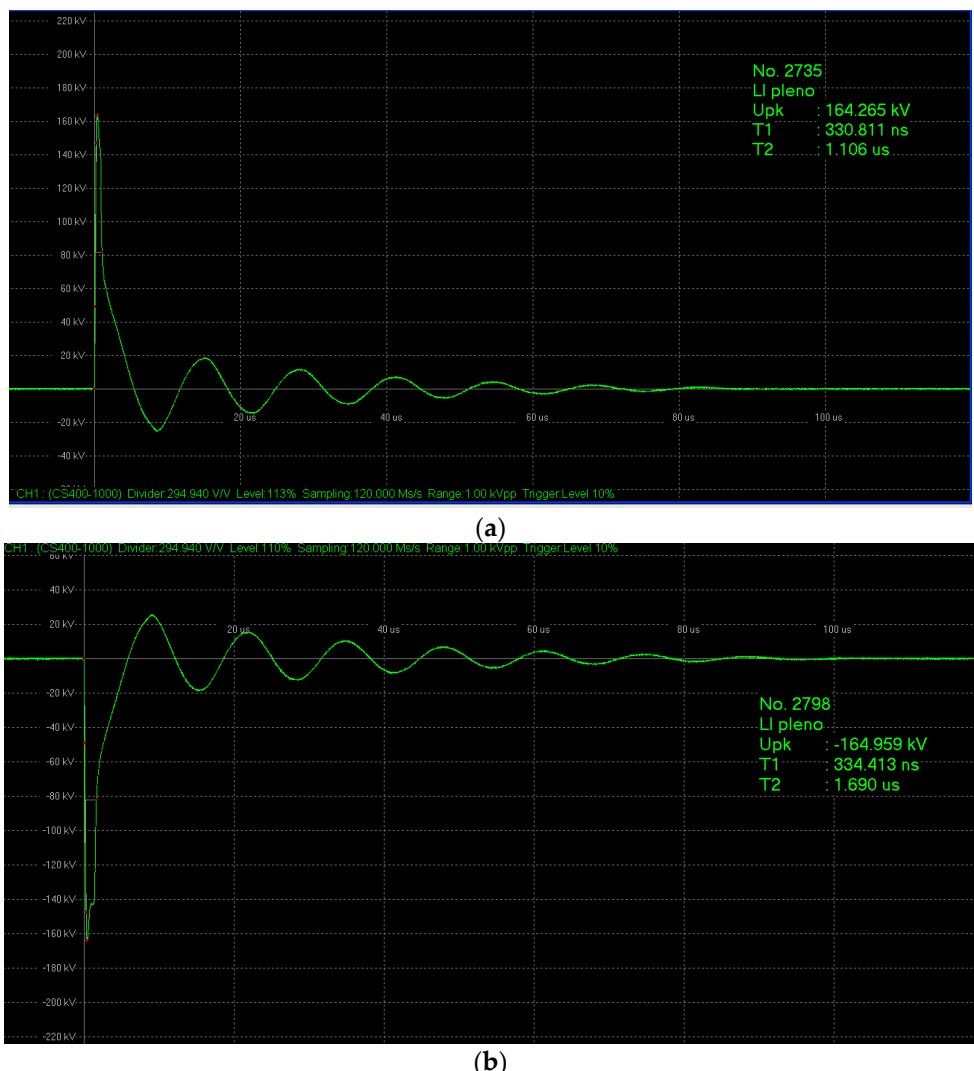

(**a**)

(**b**)

**Figure 7.** Wave shapes after impulse discharge on dunite sample: (**a**) positive polarity; (**b**) negative polarity.

2.2.2. HVEP Test Cell

In order to correctly apply generated pulses to the mineral sample, a test cell was developed that was to be attached to the Marx pulse generator, following the scheme proposed in [21]. Because the peak voltage values could reach hundreds of kV, the insulator definition, electrode configuration and distances among live elements and grounded elements were critical.

The basis of the test cell was an inox steel vessel acting as the grounded electrode. This vessel has a high-density polyethylene (HDPE) shell inside it that acts as an insulator. The active electrode is also embedded in HDPE and is supported by 3D printed parts that stabilize the whole (Figures 8 and 9), so a flat-tip electrode configuration is defined.

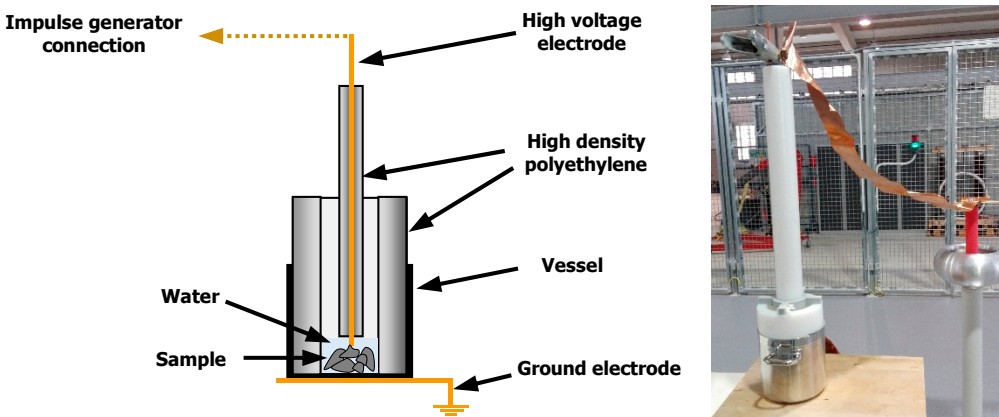

**Figure 8.** (**Left**): Test cell diagram. (**Right**): Test cell connected to the impulse generator.

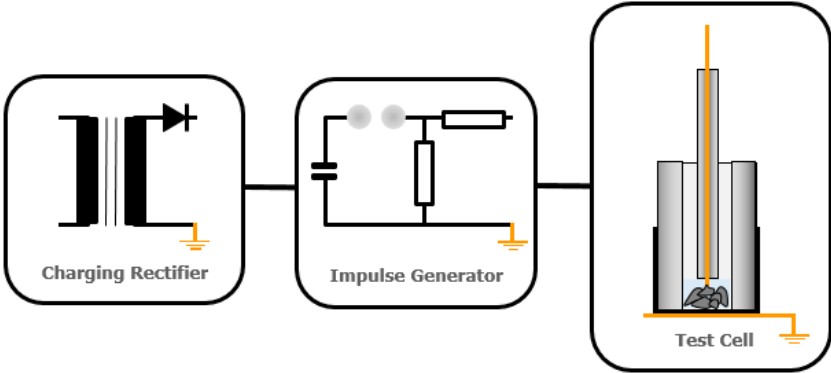

**Figure 9.** Diagram of the pulse generator and coupled test cell.

The mineral sample and the dielectric liquid are placed at the bottom of the steel vessel, which, in turn, rests on a grounded copper sheet. The active electrode, connected to the pulse generator output, comprises a copper rod that comes into contact with the sample. The HDPE cylindrical pieces guarantee that no electric arcs are formed outside the sample volume. With this electrodes configuration and the expected voltage values, the electrode distance was estimated at 25 mm; this value is in line with the values reported in [11,20–22], within the interval 20–40 mm.

### 2.2.3. HVEP Test Procedure

The tests were carried out on the pulse generator, applying high-voltage electrical pulses. At each monosize, a total of fourteen tests was performed, seven tests with positive polarity and seven more with negative polarity, in order to establish the possible influence of polarity on the degree of fragmentation of the sample. At each polarity, four samples were tested with one, two, three and four pulses, respectively. The three remaining samples were tested using five pulses to determine the test's repeatability on the final PSD.

After each test, the collected sample was dried to remove the distilled water used as a dielectric medium and sieved to obtain the PSD.

### 2.2.4. Mathematical Model

A mathematical model that describes the effect of electrofragmentation on PSD is proposed, based on an adaptation of the Cumulative Kinetic Model [23,24] into a discontinuous process, as expressed in Equation (1).

$$W_{(x,i)} = W_{(x,f)} \cdot e^{-k \cdot i} \tag{1}$$

wherein:

$W_{(x,i)}$ is the cumulative oversize of size class $x$ after $i$ pulses;
$W_{(x,f)}$ is the cumulative oversize of size class $x$ in the feed;
$k$ is the breakage rate parameter.

The relationship between the breakage rate parameter and the particle size is shown in Equation (2):

$$k = a \cdot x^b \tag{2}$$

where $a$ and $b$ can be determined experimentally. Accordingly, once one has defined the model parameters, the electrofragmentation product PSD after $i$ pulses can be obtained from the feed PSD using Equation (3).

$$W_{(x,i)} = W_{(x,0)} \cdot e^{-a \cdot x^b \cdot i} \tag{3}$$

The $k$ value is determined for each monosize after taking logarithms at Equation (1):

$$\ln(W_{(x,i)}) = \ln(W_{(x,0)}) - k \cdot i \Rightarrow \ln(W_{(x,i)}) - \ln(W_{(x,0)}) = k \cdot i \tag{4}$$

Once one has obtained $k$ values for each monosize, an additional linear regression can be performed to calculate $a$ and $b$, according to Equation (5).

$$\ln(k) = \ln(a) + b \cdot \ln(x) \tag{5}$$

## 3. Results and Discussion

Tables S1–S10 show the results of the 70 impulse tests performed on different monosizes, with positive and negative polarity, including the three five-pulse replicas.

Figure 10 compares the PSD values (cumulative oversize) in the monosize 5000/3350 μm case when using different polarities. In the case of no influence of the polarity, values should be randomly spread following the diagonal line. However, in this case, plotted points are located above the diagonal line due to the cumulative oversize value being higher in the case of positive polarity; this means that the comminution effect is higher in the case of negative polarity. This monosize shows the same behavior in the case of one to four pulses, while in the case of five pulses, values almost fit the diagonal, thus meaning that polarity does not influence the PSD after five pulses.

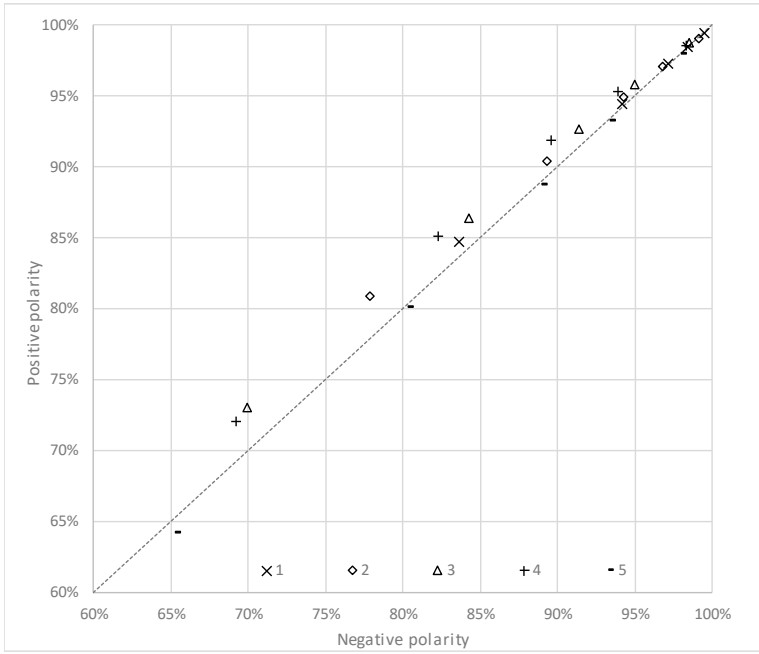

**Figure 10.** Product PSD after a different number of pulses (feed monosize 5000/3350 μm) and different polarity.

The same analysis was performed with the rest of the monosizes. Figure 11 shows the result in the 3350/2000 μm size interval case, which shows an opposite behaviour from the previous monosize. In this case, the positive polarity seems to produce a more intense comminution effect in the case of one to four pulses, while again, in the case of five pulses, the polarity seems not to influence the PSD. On the other hand, with monosizes 2000/1000 μm, 1000/500 μm, and 500/125 μm (Figures 12–14), the results suggest that the polarity does not influence the comminution effect. From these results, the influence of the polarity cannot be concluded; however, under certain conditions, the results show that a specific polarity could improve the comminution effect in the electrofragmentation device.

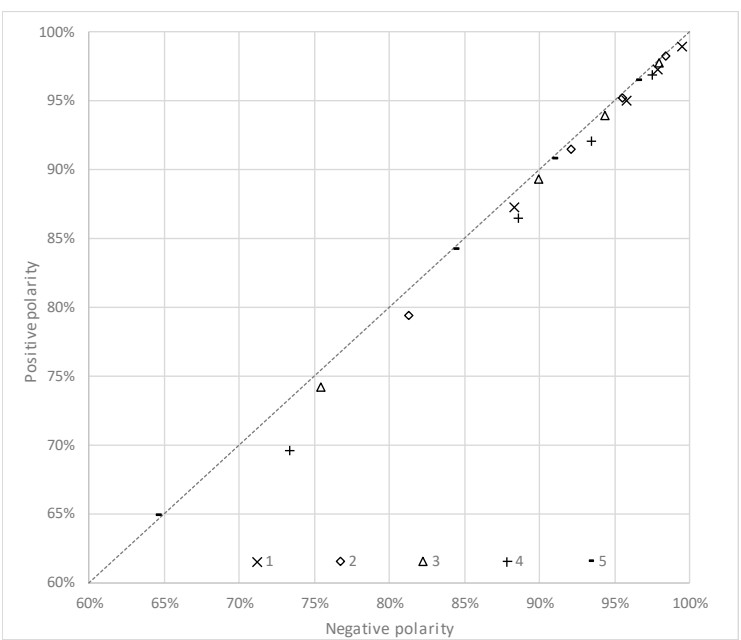

**Figure 11.** Product PSD after a different number of pulses (feed monosize 3350/2000 μm) and different polarity.

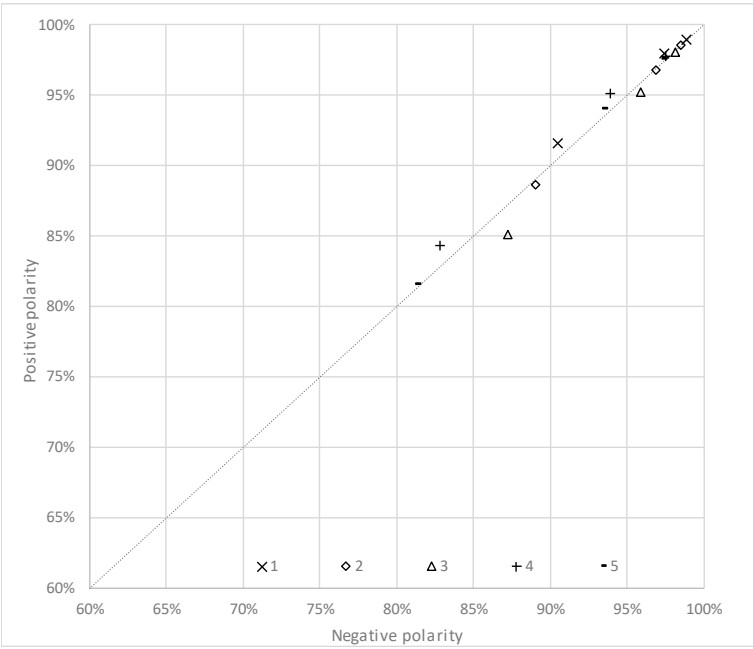

**Figure 12.** Product PSD after a different number of pulses (feed monosize 2000/1000 μm) and different polarity.

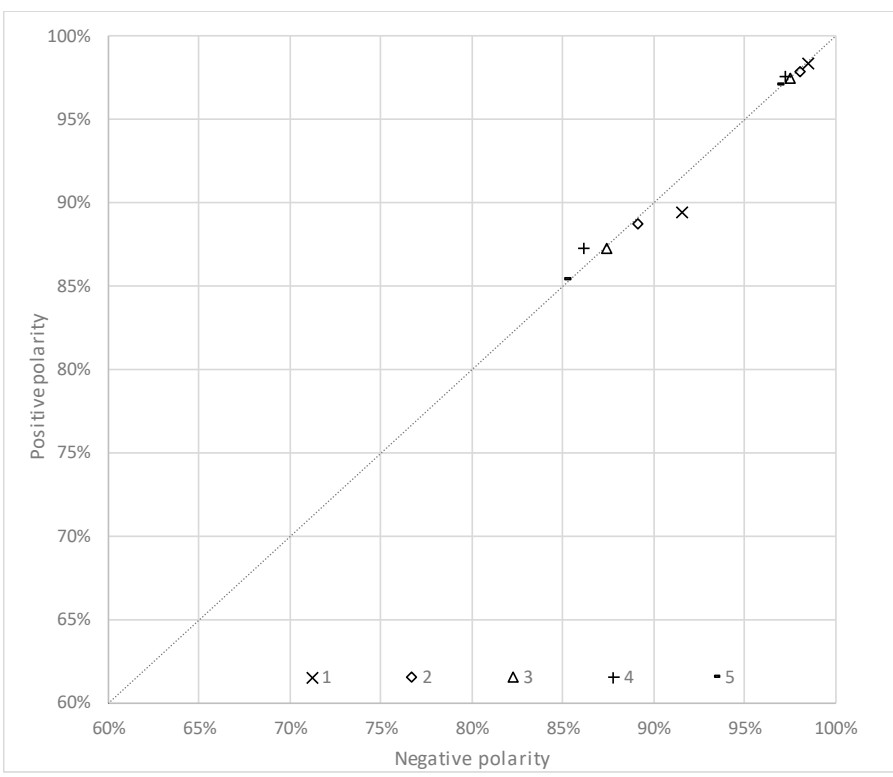

**Figure 13.** Product PSD after a different number of pulses (feed monosize 1000/500 μm) and different polarity.

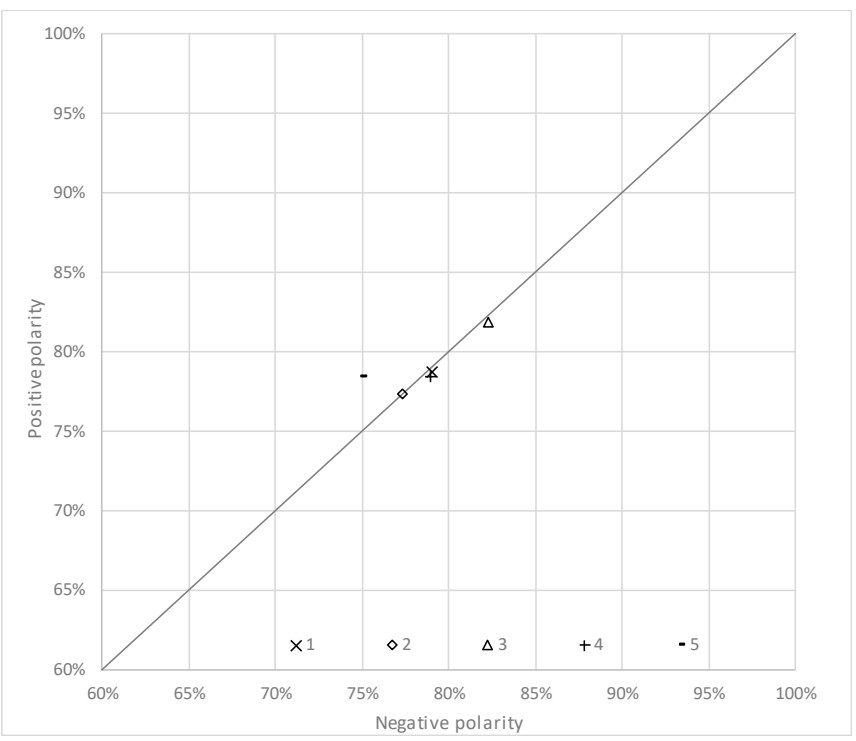

**Figure 14.** Product PSD after a different number of pulses (feed monosize 500/125 μm) and different polarity.

Regarding the comminution modeling, from data gathered in Tables S1–S10 and Equations (4) and (5), the proposed model parameters can be calculated, again for each polarity. Table 4 shows the results of *a* and *b* parameters and the correlation coefficient value

obtained in Equation (5) for linear regression. According to the $R^2$ values, both polarities show a better fit at coarser monosizes, with very similar values.

**Table 4.** Model parameter values calculated.

| Monosize | Negative Polarity | | | Positive Polarity | | |
|---|---|---|---|---|---|---|
| (µm) | *a* | *b* | $R^2$ | *a* | *b* | $R^2$ |
| 5000/3350 | 0.00006 | 0.85365 | 0.99730 | 0.00004 | 0.90009 | 0.99940 |
| 3350/2000 | 0.00014 | 0.78421 | 0.97140 | 0.00017 | 0.77116 | 0.97210 |
| 2000/1000 | 0.00019 | 0.63905 | 0.88830 | 0.00010 | 0.72087 | 0.88710 |
| 1000/500 | 0.00076 | 0.41263 | 0.68810 | 0.00057 | 0.46036 | 0.68810 |

The parameter values shown in Table 4 were calculated by considering replica 1 at five pulses, in order to compare the model's estimated PSD with the remaining replicas. Table 5 gathers the results obtained with both polarities, in the case of the 5000/3350 µm monosize; these results are also plotted in Figure 15. Tables S11–S13 in the Supplementary Material gather the results of the other monosizes.

**Table 5.** PSD values (modeled and real), feed 5000/3350 µm monosize, five pulses.

| Size | Negative Polarity | | | Positive Polarity | | |
|---|---|---|---|---|---|---|
| (µm) | Model | Replica 2 | Replica 3 | Model | Replica 2 | Replica 3 |
| 3350 | 61.54% | 65.94% | 63.61% | 63.24% | 61.95% | 64.68% |
| 2000 | 77.29% | 80.73% | 79.70% | 78.62% | 79.42% | 79.74% |
| 1000 | 87.03% | 89.31% | 88.67% | 88.17% | 88.20% | 88.42% |
| 500 | 92.59% | 93.69% | 93.12% | 93.39% | 92.88% | 93.04% |
| 125 | 97.62% | 98.09% | 97.85% | 97.97% | 97.81% | 97.92% |

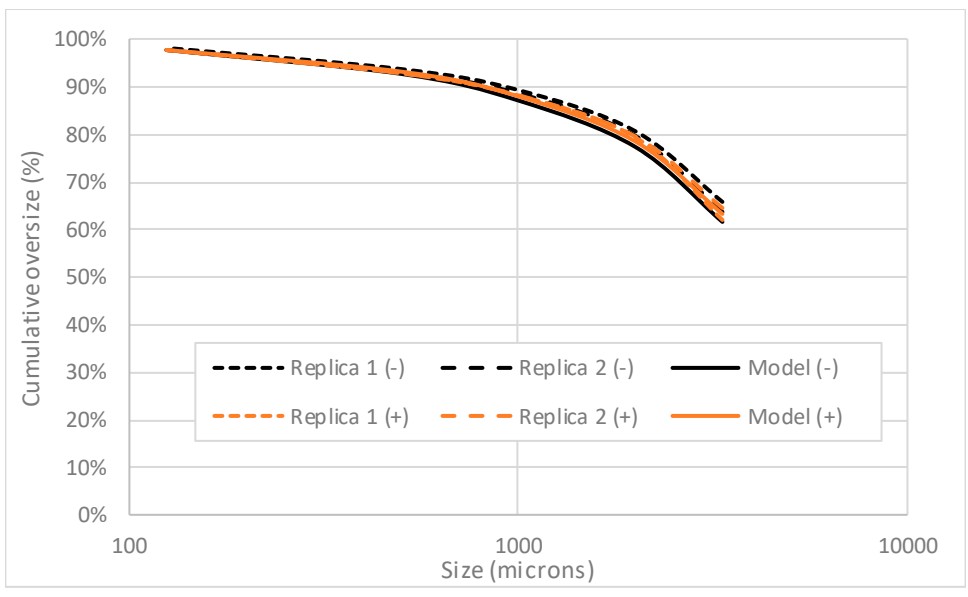

**Figure 15.** Product PSD after five pulses, monosize 5000/3350 µm.

In order to analyses the results, a first comparison was made between replicas 2 and 3. Subsequently, a second comparison was performed between the modeled PSD values and the average distribution obtained from replicas 2 and 3 (labeled as real). Model deviation had a relative error lower than 2%, which was even lower than 0.5% at finer monosizes. The F-test values are shown in Table 6 for all monosizes and both polarities.

**Table 6.** F-test values obtained in the comparisons performed.

| Monosize | Negative Polarity | | Positive Polarity | |
|---|---|---|---|---|
| (µm) | Among Replicas | Model/Real | Among Replicas | Model/Real |
| 5000/3350 | 0.9088 | 0.8724 | 0.8876 | 0.9777 |
| 3350/2000 | 0.9974 | 0.9789 | 0.9995 | 0.9561 |
| 2000/1000 | 0.9833 | 0.9802 | 0.9943 | 0.9825 |
| 1000/500 | 0.9928 | 0.9919 | 0.9920 | 0.9836 |
| 500/125 | 0.9891 | 0.8543 | 0.9218 | 0.9868 |

According to the results shown in Figure 15, in general terms, the proposed model achieves a good fitting of PSD after five pulses, with a slightly better result in the case of positive polarity; this can also be deduced from the F-values shown in Table 6, obtaining a value of 0.9777 in the case of positive polarity, which is higher than the value obtained in the case of negative polarity, 0.8724. Further research must be performed with different ores and pulse conditions to define the influence of pulse polarity.

## 4. Conclusions

From the results obtained in this research, the following conclusions can be highlighted:

- With a monosize 5000/3500 µm, a negative polarity achieved a better comminution effect, while with the monosize 3350/2000 µm, a positive polarity achieved better performance. In finer monosizes, the polarity effect was not conclusive. Accordingly, the influence of the polarity on the electrofragmentation effect cannot be concluded, and further studies should be performed;
- The proposed model can achieve a good prediction of the electrofragmentation product PSD, after a given number of impulses. The results for both polarities were similar, with a slightly better result in the case of positive polarity.

**Supplementary Materials:** The following supporting information can be downloaded at: https://www.mdpi.com/article/10.3390/met12030494/s1, Table S1: Results obtained with monosize 5000/3350, negative polarity; Table S2. Results obtained with monosize 5000/3350, positive polarity; Table S3. Results obtained with monosize 3350/2000, negative polarity; Table S4. Results obtained with monosize 3350/2000, positive polarity; Table S5. Results obtained with monosize 2000/1000, negative polarity; Table S6. Results obtained with monosize 2000/1000, positive polarity; Table S7. Results obtained with monosize 1000/500, negative polarity; Table S8. Results obtained with monosize 1000/500, positive polarity; Table S9. Results obtained with monosize 500/125, negative polarity; Table S10. Results obtained with monosize 500/125, positive polarity; Table S11: PSD values (modeled and real), feed 3350/2000 monosize; Table S12: PSD values (modeled and real), feed 2000/1000 monosize; Table S13: PSD values (modeled and real), feed 1000/500 monosize.

**Author Contributions:** Conceptualization, A.R.L., J.M.M.-A. and M.G.M.; methodology, J.M.M.-A., M.G.M. and F.J.P.; software, A.R.L. and A.D.; validation, F.J.P. and M.G.M.; formal analysis and investigation, A.R.L., F.J.P. and M.G.M.; resources, J.M.M.-A. and M.G.M.; writing—original draft preparation, A.R.L., A.D. and F.J.P.; writing—review and editing, J.M.M.-A. and M.G.M.; supervision, J.M.M.-A. and M.G.M. All authors have read and agreed to the published version of the manuscript.

**Funding:** This research was partially funded by the Spanish Ministry of Economy and Competitiveness, under project DPI2017-83804-R.

**Institutional Review Board Statement:** Not applicable.

**Informed Consent Statement:** Not applicable.

**Data Availability Statement:** Not applicable.

**Conflicts of Interest:** The authors declare no conflict of interest.

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
