# Peer review of "Study of Comminution Kinetics in an Electrofragmentation Lab-Scale Device"

_metals, doi:10.3390/met12030494_

Round 1
Reviewer 1 Report
The draft expresses just the results of PSDs obtained in the experiments and little kinetic analysis and scientific consideration. Kinetic analysis, shown in the title, should be carried out and the scientific reasons for the results should be clarified, by comparing the past results. I would also like to tell you that the PSD should written as cumulative "under" size distribution not "oversize", which is the suggestion of the ISO.
Author Response
R: The draft expresses just the results of PSDs obtained in the experiments and little kinetic analysis and scientific consideration. Kinetic analysis, shown in the title, should be carried out and the scientific reasons for the results should be clarified, by comparing the past results. I would also like to tell you that the PSD should written as cumulative "under" size distribution not "oversize", which is the suggestion of the ISO.
A: Authors appreciate the reviewer’s suggestions, which have been included in the revised version of the manuscript. We want to highlight that the obtention of the presented PSD’s was possible thanks to the 3-year development of both adequate device and test protocol, containing this paper the first set of results of our research available to the public. Regarding the kinetic analysis, the authors consider that providing for the first time that the Cumulative Kinetic Model (CKM) can be applied to electrofragmentation tests modelling with accuracy is worthy of publication.
Reviewer 2 Report
The main problem of this manuscript is that it is more a summary of experimental report rather an discovery. The model developed by the authors only is a description of the product size distribution, and no sound relation between product fineness and high voltage pulse treatment conditions were established. Yes equation 6 try to relate impluse number to product fineness. However, firstly the R2 value in table 4 decreases as the feed size reduces. Secondly, the relation of k value to impulse number was represented by a and b value, but no effect by voltage, electrode gap, etc was reported on the a and b value. The a and b values of different test series are indepedent to each other. This is why I consider the model in this research is a description of experimental results rather than a real discovery.
I do not recommend publishing unless the model reported can be better related to the testing conditions.
Author Response
R: The main problem of this manuscript is that it is more a summary of experimental report rather an discovery.
A: Authors appreciate the reviewer point and agree that we present mainly an experimental report. Although we developed our electro-impulse fragmentation (EIF) device, of course we did not discover this methodology. The objective of this work was to research on kinetic properties of EIF, which is very little covered by literature. So we give much value to this experimental report and are sure of the interest to the readers.
R: The model developed by the authors only is a description of the product size distribution, and no sound relation between product fineness and high voltage pulse treatment conditions were established.
A: This series of tests were aimed to research the grinding kinetics and see if CKM model, adapted to a batch process, could be applied. The EIF device conditions were fixed to values recommended by preliminary tests, as described in the paper.
R: Yes equation 6 try to relate impluse number to product fineness. However, firstly the R2 value in table 4 decreases as the feed size reduces.
A: The influence of impulse number could be foreseen, but the proposed model proved to work; even considering the R2 decrease with feed size, the fitting level can be considered adequate.
R: Secondly, the relation of k value to impulse number was represented by a and b value, but no effect by voltage, electrode gap, etc was reported on the a and b value. The a and b values of different test series are indepedent to each other. This is why I consider the model in this research is a description of experimental results rather than a real discovery.
I do not recommend publishing unless the model reported can be better related to the testing conditions.
A: Authors agree that much more tests should be made to offer a “real discovery”, as stated by the reviewer; however, please consider that the CKM approach has more than 40 years of research and validation in grinding, and no correlation has been stated between a and b parameters and specific tumbling mill parameters. This fact does not diminish the value of the research in tumbling mills and, accordingly, our research on EIF approach.
Reviewer 3 Report
Dear Author,
The reviewer questions were answered, and the changes were included in the manuscript.
Regards
Author Response
Authors appreciate the help provided by the reviewer to improve the manuscript
Reviewer 4 Report
This study is very good original research paper, which proposes a mathematical model to predict product PSD from a novel comminution device called “high voltage electric pulses device” after one or more electric pulses under specific working conditions.
Positive points:
Subject matter is within the scope of the journal,
Title accurately reflects content,
language is grammatically correct,
abstract is clear and adequate,
Presentation and illustrations are good and conform to acceptable standards,
References are appropriate.
Issues that need attention:
- Perspectives for future works need to be more detailed.
- Discuss whether predicted and measured shape values were found close to each other?
- Absolute and relative error values can be calculated and discussed for experimental and predicted values.
But, the material studied in this manuscript is dunite rock ore not metal, since the topics covered in the journal of Metals are all kinds of metals. Therefore, I recommend to submit this valuable manuscript to another appropriate journal.
Author Response
R: This study is very good original research paper, which proposes a mathematical model to predict product PSD from a novel comminution device called “high voltage electric pulses device” after one or more electric pulses under specific working conditions.
Positive points:
Subject matter is within the scope of the journal,
Title accurately reflects content,
language is grammatically correct,
abstract is clear and adequate,
Presentation and illustrations are good and conform to acceptable standards,
References are appropriate.
A: Authors appreciate the detailed revision and analysis performed.
R: Issues that need attention:
- Perspectives for future works need to be more detailed.
A: Thank you, this suggestion was addressed in the revised version of the manuscript
R: 2. Discuss whether predicted and measured shape values were found close to each other?
A: Thank you, this was included in the discussion in the revised version of the manuscript
R: 3. Absolute and relative error values can be calculated and discussed for experimental and predicted values.
A: Thank you, this was included in the discussion in the revised version of the manuscript
R: But, the material studied in this manuscript is dunite rock ore not metal, since the topics covered in the journal of Metals are all kinds of metals. Therefore, I recommend to submit this valuable manuscript to another appropriate journal.
A: Thank you for the comment. Pasek Minerals, the company that provided the rock samples, has developed a process to extract Magnesium from dunite rock; so dunite rock is, in fact, a metal ore; some comments on this have been included in the revised version of the manuscript. However, the presented methodology can be applied to other metal ores without restrictions.
Round 2
Reviewer 1 Report
My decision, unfortunately, could not be changed because the draft expressed just experimental results and involve little scientific consideration, and should not be published as an original paper.Reviewer 2 Report
The authors had responsed well the problems and the manuscript has been sufficiently improved to warrant publication in Metals.
Reviewer 4 Report
The authors have revised based on the reviewer’s comments and now paper has a sufficient quality for the publication.
This study can be accepted as it is.